# Dynamic Changes in Human Milk Oligosaccharides in Chinese Population: A Systematic Review and Meta-Analysis

**DOI:** 10.3390/nu13092912

**Published:** 2021-08-24

**Authors:** Yalin Zhou, Han Sun, Kaifeng Li, Chengdong Zheng, Mengnan Ju, Ying Lyu, Runlong Zhao, Wenqing Wang, Wei Zhang, Yajun Xu, Shilong Jiang

**Affiliations:** 1Department of Nutrition and Food Hygiene, School of Public Health, Peking University, Xueyuan Road 38, Haidian, Beijing 100083, China; zylyingyang@163.com (Y.Z.); lybjmu@126.com (Y.L.); 18210192169@163.com (R.Z.); 2PKUHSC-China Feihe Joint Research Institute of Nutrition and Healthy Lifespan Development, Xueyuan Road 38, Haidian, Beijing 100083, China; sunhan@feihe.com (H.S.); likaifeng@feihe.com (K.L.); zhengchengdong@feihe.com (C.Z.); jumengnan@feihe.com (M.J.); wangwenqing@feihe.com (W.W.); zhangwei1@feihe.com (W.Z.); 3Innovation Center, Nutrition and Metabolism Research Division, Heilongjiang Feihe Dairy Co., Ltd., C-16, 10A Jiuxianqiao Rd., Chaoyang, Beijing 100015, China; 4Beijing Key Laboratory of Toxicological Research and Risk Assessment for Food Safety, Peking University, Xueyuan Road 38, Haidian, Beijing 100083, China

**Keywords:** breast milk, lactation, longitudinal changes, sialic acid, fucose

## Abstract

The aim of this systematic review was to summarize concentrations of human milk oligosaccharides (HMOs) in the Chinese population. We searched articles originally published in both Chinese and English. When compiling data, lactation was categorized into five stages. We found that 6′-sialyllactose, lacto-*N*-tetraose, and lacto-*N*-neotetraose decreased over lactation. Conversely, 3′-fucosyllactose increased over lactation. Our study represents the first systematic review to summarize HMO concentrations in Chinese population. Our findings not only provide data on HMO profiles in Chinese population but suggest future directions in the study of the metabolism of HMOs.

## 1. Introduction

Human milk represents the optimal source of nutrition for neonatal growth, development, and health. Currently, there are gaps between breastfed and formula-fed infants. For example, when compared with breastfed infants, formula-fed infants gain excessive weight, possess lower abundance of *Bifidobacteria* in the gut, have higher incidences of infections, etc. [1]. Furthermore, the differences between breastfeeding and formula feeding could prolong to later life, especially in cognitive development and risks of metabolic disorders [2]. The advantages of breastfeeding are mainly attributed to human milk constituents. 

Human milk oligosaccharides (HMOs) are the fourth largest group of nutrients in human milk after water, lactose, and lipids [3]. Concentrations of total HMOs range from 5–25 g/L in human milk [4,5,6]. Although HMOs are not a significant source of metabolic energy or building blocks of tissues such as lactose, lipids, and protein, there are several possible mechanisms via which HMOs play vital roles in infant health. First, HMOs nourish probiotics in the gut as a carbon source [7]. Second, HMOs block the adhesion of bacterial and viral pathogens to the surface of intestinal epithelial cells [8,9]. Third, HMOs directly interact with cells of the body and modulate cellular functions [10]. Because intact HMOs can enter the circulation, HMOs can theoretically interact with all cells in the body [1]. Fourth, HMOs can donate fucose and sialic acid (*N*-acetylneuraminic acid) moieties in the glycosylation of proteins and lipids [11]. 

HMOs are highly diverse in structures. Currently, more than 200 HMOs have been characterized [6]. Nevertheless, structures of HMOs generally share a common blueprint [12]: (1) the backbones of HMOs all contain lactose at the reducing end; (2) lactose at the reducing end can be elongated by adding glucose (Glc), galactose (Gal), and *N*-acetylglucosamine (GlcNAc); (3) backbones of HMOs can be modified by adding fucose or sialic acid moieties. 

Interindividual differences in HMO profiles due to genetic factors have been demonstrated. The α1-2-fucosyltransferase (FUT2; encoded by *Se* gene) is involved in the fucosylation in the synthesis of α1-2-fucosylated HMOs such as 2’-fucosyllactose (2′-FL) and lacto-*N*-fucopentaose I (LNPF I) [4,13]. Single-nucleotide polymorphisms (SNPs) in *Se* may cause dramatically lower concentrations of α1-2-fucosylated glycans in human milk [14]. Similarly, α1-3/4-fucosyltransferase (FUT3; encoded by *Le* gene) participates in the synthesis of α1-4-fucosylated HMOs such as lacto-*N*-fucopentaose II (LNFP II) [4,13]. Accordingly, SNPs in *Le* may lead to lower levels of α1-4-fucosylated HMOs in human milk [15]. It should be noted that more than 200 glycosyltransferases have been identified in the human genome [16]. Moreover, other families of enzymes are involved in the biogenesis of HMOs [17]. Therefore, it is possible that HMOs in addition to α1-2-fucosylated and α1-4-fucosylated glycans may display interindividual variations. 

Human milk from different populations may exhibit different typical HMO profiles due to genetic factors [18]. For example, frequencies of SNPs in *Se* gene were found different among populations [19,20]. Previously, Thurl et al. attempted to establish representative HMO profiles by conducting a systematic review and compiling data from 15 countries and regions (except for the Mainland of China) [3]. Additionally, they identified that genetic factors, as well as lactation stage, affect HMO concentrations. To our best knowledge, the study by Thurl et al. remains the only systematic review on HMO concentrations to date. Unfortunately, none of the included studies reported HMO data from the mainland of China. 

China has more than 20% of the world population. In recent years, HMO studies in the Chinese population have been published in both English and Chinese, which are not readily available to non-Chinese scholars and policymakers. In our initial literature search, we found that small sample sizes and limited locations of sample collection were common in the original studies that reported HMOs in Chinese human milk, making it impossible to conclude typical HMO profile in Chinese population from any of the studies. Accordingly, the aim of this study was to compile HMO concentrations in Chinese population by a systematic review. We also attempted to interrogate the influences of lactation stages and genetic factors on HMO concentrations. The findings of our study could potentially enhance the knowledge of lactation biology, improve infant feeding practices, and shed light on the metabolism of HMOs in the infant body. 

## 2. Methods

### 2.1. Literature Selection

The guidelines in Preferred Reporting Items for Systematic Review and Meta-Analysis Protocols (PRISMA-p) were followed [21]. Briefly, for the literature published in English, the databases Web of Science, PubMed, and ScienceDirect were searched using the formula “(human OR breast) AND (milk) AND (oligosaccharide*) AND (composition OR concentration OR content* OR quantification OR profile* OR amount) AND (China OR Chinese)”. For the literature published in Chinese, the databases China National Knowledge Infrastructure (CNKI) (http://cnki.net, accessed on 25 May 2020), Wanfang (http://www.wanfangdata.com.cn/index.html, accessed on 25 May 2020), and Chongqing VIP Information (http://qikan.cqvip.com, accessed on 25 May 2020) were searched, and the formula was optimized to “(human milk OR breast milk) AND (oligosaccharide)” to adapt to the style of the Chinese language. Literature searching was completed in May 2020. The titles and abstracts of all hits were checked, and the obviously unrelated articles were removed. The full texts of the remaining articles were examined by applying the inclusion and exclusion criteria listed in the PICOS table (Appendix A). Quality of included studies was assessed as previously described [22]. The literature selection was performed independently by two investigators (Y.Z. and H.S.). Discrepancies were addressed in the presence of a third investigator (K.L.). 

### 2.2. Data Mining and Processing

To compile data, lactation was divided into five stages: 1–7, 8–14, 15–60, 61–120, and beyond 121 postnatal days. Means and standard deviations (SDs) of HMO concentrations were extracted. Units were all converted to mg/L. The human milk density was 1.03 g/mL in the conversions [23]. Data mining and processing was performed independently by two investigators (Y.Z. and H.S.). Discrepancies were addressed in the presence of a third investigator (K.L.). Nomenclatures were adapted from Bode (2012) [1].

### 2.3. Statistical Analysis

Weighted means, SDs, and standard errors (SEs) of HMOs in each lactation stage were calculated as described [24]. The 95% CIs of each HMO across lactation stages were estimated as weighted mean ± t × SE, where t follows a t-distribution with degrees of freedom equal to the group sample size minus 1. Cochran’s Q statistic of a fixed effect model, which, under the null hypothesis of no subgroup (namely, lactation stage), heterogeneity follows a *Χ*^2^-distribution with degrees of freedom equal to number of subgroup minus 1, was calculated [25]. A random effect model was used when significant heterogeneity was observed using the fixed model. Higgins and Thompson’s *I*^2^ was derived according to the *Q* statistic [26]. Figures were generated by using ggplot2 R package [27]. All the statistical analyses were performed using R (version 4.0.3) [28]. Forrest plots were generated to compare HMO concentrations of included studies using Stata/SE 14.0 (StataCorp LLC, College station, TX, USA). Publication biases were estimated by plotting mean concentration of HMO against its corresponding SE of each study included. 

## 3. Results

The article selection process is summarized in Figure 1. Included studies [4,14,29,30,31,32,33,34] were listed (Table 1) and assessed (Appendix A). Articles were published from 2009 to 2020. Publication biases of included studies were visualized by the funnel plots (Appendix A). Concentrations of 14 structures of HMOs were reported in the included articles (Appendix A). Among the investigated HMOs, eight were fucosylated, four were sialylated, and two were not modified by fucose or sialic acid. Lengths of the included glycans range from 3–6 sugar residues with four trisaccharides, two tetrasaccharides, five pentasaccharides, and three hexasaccharide. One study reported SNPs in *Se*. None of the included studies directly investigated Lewis types or SNPs in *Le*. 

Significant between-study heterogeneities were observed for 2‘-FL, 3′-FL, 6′-SL, and 3′-SL (Figure 2). This remained true for other HMOs included in our study (Appendix A). To characterize the longitudinal pattern of each HMO during lactation stages, subgroup analysis was carried out using random effect model considering substantial between-study homogeneities being observed (Table 2 and Figure 3). 

Fucosylated HMOs exhibited distinct patterns of changes over lactation. The concentration of 2′-FL in the 1–7 postnatal days was higher than that at all following lactation stages (Figure 3). The concentration of 2′-FL in the 8–14 postnatal days was about 60% of that in the 1–7 postnatal days (Table 2). Nevertheless, longitudinal changes of 2′-FL over lactation was not significant according to subgroup heterogeneity analysis using a random effect model (Table 2). (*Q* = 7.0, *df* = 4, *p* = 0.137). By contrast, the concentrations of 3′-fucosyllactose (3′-FL) significantly increased (*Q* = 104.5, *df* = 4, *p* < 0.001) with the progression of lactation according to the subgroup heterogeneity analysis (Figure 3 and Table 2). The sum of 2′-FL and 3′-FL remained constant over lactation (*p* = 0.538). Additionally, the concentration of lacto-*N*-difucohexaose II (LNDFH II) in the 61–120 postnatal days was significantly (*Q* = 290.7, *df* = 3, *p* < 0.001) higher than that at the preceding stages (Figure 3 and Table 2). There was no clear pattern of longitudinal changes in LNFP I, LNFP II, lacto-*N*-fucopentaose III (LNFP III), and lacto-*N*-fucopentaose V (LNFP V) (Figure 3). In the first 120 postnatal days, 2′-FL was the dominant fucosylated HMO (Figure 4). In the postnatal days beyond 121 days, 3′-FL was the most abundant fucosylated HMO (Figure 4). The concentration of 2′-FL was higher than 900 mg/L during the entire lactation (Table 2).

## 4. Discussion

Our study provides a summary of 14 HMOs in Chinese population by using a defined literature search and selection process. We compiled data from original research articles that were published in both English and Chinese. Our study demonstrates that HMO concentrations exhibit dynamic changes during lactation. The concentrations of 6′-SL, LNT, and LNnT decline over lactation. Conversely, the concentration of 3′-FL increases with the progression of lactation. The longitudinal changes in LNFP I, LNFP II, LNFP V, and 3′-SL are not significant. Our study is the first systematic review to establish reference HMO profile in Chinese population. 

Unlike previous studies on HMO content from varied populations in distinct countries, our study focused on Chinese mothers. Although variations in detection methods and targeted population from studies to studies limit the comparability of the data, the dynamic change in HMOs along the lactation period may comply with a general trend.

We calculated the total concentration of 14 kinds of HMOs and observed the downward trend with the progression of lactation from the holistic perspective (Figure 4). The findings were supported by previous studies. Many studies demonstrated that total concentrations of interest HMOs were the highest in the first week of lactation and decreased thereafter [4,5]. This result is in accordance with the overall dynamic changes in total concentration of all HMOs along the lactation period. Although it is not possible to measure all the HMOs in human milk, it has been reported that the total level of HMOs in colostrum is estimated to be the highest at 9–22 g/L and declines slightly in transitional milk (8–19g/L at postnatal 8–15 days), followed by a gradual decrease in mature milk [35]. However, it is worth noting that the total content of 14 HMOs showed a sharp increase in the 61–120 postnatal days, which is likely attributed to the dramatic rise in the concentration of DSLNT. This may be explained by the scant number of studies on DSLNT and the limited sample population involved. There are insufficient basic data available on the HMO content of lactating mothers in China, implying that sufficient caution should be taken when drawing conclusions.

2′-FL is the most abundant HMO whose biological significance includes modulating immune functions, nourishing gut prebiotics, and providing fucose moieties for the glycosylation of biomolecules such as proteins and lipids [1]. Although not significantly, the concentration of 2′-FL declined gradually along with the lactation course, which is consistent with almost all studies [11,35]. 3′-FL is another neutral fucosylated HMO. Differing from 2′-FL, the level of 3′-FL rose significantly and may have compensated for 2′-FL. It is possible that the secretion of 3′-FL, an isomer of 2′-FL, is upregulated with the progression of lactation to compensate for 2′-FL. In our study, although 2′-FL and 3′-FL concentrations displayed opposite trends, the sum of 2′-FL and 3′-FL remained constant after the first week of lactation. In another study, researchers reviewed the HMO concentration among populations from different countries and observed an increasing trend of 3′-FL and a negative correlation between the production of 2′-FL and 3′-FL [35]. These observations implied the possibility that 3′-FL may compensate for 2′-FL in human milk from non-secretor mothers whose 2′-FL is dramatically lower than that from secretor mothers. Subsequently, another question was raised regarding the potential compensatory mechanisms of 2′-FL in non-secretor mothers. Therefore, in the future, studies that directly compare 2′-FL with 3′-FL in their functions and metabolism in infants are necessary to formally evaluate the compensatory mechanism of 2′-FL by 3′-FL. 

Despite no significance, other neutral fucosylated HMOs such as LNFP I and LNFP V decreased during the course of lactation, which is in accordance with findings reported in previous studies [3,11,35]. LNnT and LNT, neutral non-fucosylated HMOs, declined throughout the lactation, and the changes reached the significant level. Sumiyoshi [36] also reported the similar findings among Japanese population. Additionally, LNnT and LNT showed parallel dynamic changes among population from North America, South America, and Europe reported by McGuire [11].

In our study, the concentrations of two major sialylated HMOs (6′-SL and 3′-SL) showed a downward trend across lactation, while the decrease in 3′-SL did not reach a significant level, unlike findings reported in a previous study [37]. 6′-SL predominated in the early period of lactation (<15 days) and declined with the progression of lactation. Beyond 120 days, the concentrations of 6′-SL were comparable to those of 3′-SL. In line with our findings, it was proposed that 6′-SL occupies the main position among sialylated HMOs at early stages of lactation (<3 months), and the concentration of 3′-SL is higher at 4–8 months [3,35,37].

DSLNT is another major acidic HMO. In several previous studies, the level of DSLNT declined from colostrum to transitional milk to mature milk, but the longitudinal changes were not significant [37,38]. Conversely, unlike the findings reported, we found that the concentration of DSLNT in the 61–120 day stage rose dramatically to at least 1.5 times that at preceding stages. We acknowledge that the results are debatable because of the scant literature and small sample size included. It is advisable for us to draw a conclusion with an abundance of caution. The sialylated oligosaccharides may have an important role in early postnatal cognitive development. It should be highlighted that each DSLNT molecule contains two sialic acid moieties, whereas 6′-SL and 3′-SL include one moiety. In addition, existing evidence underpins a significant relationship between fucosylated/sialylated HMOs in maternal milk and offspring blood [39,40]. Although sialic acid moieties widely exist in body tissues, their concentration in the gray matter of the central nervous system is more than 10 times higher than that in all other organs [41]. Sialic acid is linked with the interface between neurons (synapse), where it engages in the neuron signal transduction [42]. Although infants are born with almost all neurons, the development of synapses continues into postnatal life [43]. The velocity of synaptogenesis peaks at the age of 3 months [44]. Interestingly, 3 months is also the time when the concentration of 3′-SL becomes comparable to 6′-SL. Together, it is likely that the predominant sialylated HMOs depend on the lactation stages and play varied roles in cognitive development by providing sialic acid moieties. 

The content and composition of HMOs in milk are affected by multiple factors including environmental and maternal factors such as secretor status, lactation stages, maternal age, physical status, and diet. Some previous systematic reviewed data from different countries all around the world and investigated the influencing factors related to the concentration of HMOs. For example, there was a systematic review on data from 15 countries and reporting 33 structures of HMOs [3]. The study had the advantage of separating milk from secretor and non-secretor mothers, as well as analyzing the differences between milk from mothers delivered at term and preterm. Regretfully, in our study, only one included article directly sequenced SNPs in *Se*, and none of the studies sequenced SNPs in *Le* [14]. It is also possible to predict secretor status by calculating the ratios of 2′-FL to 3′-FL [40]. However, the prediction of secretor status is dependent on the availability of HMO concentrations from individual mothers, which was available only in the study by Chen et al. [30]. In that study, three out of five mothers were secretors [30]. Mean concentrations of 2′-FL and 3′-6′-FL were 3770.75 mg/L and 227.98 mg/L, respectively. In non-secretor mothers, mean concentrations of 2′-FL and 3′-FL were 2781.02 mg/L and 555.46 mg/L, respectively. In addition, another systematic review summarized the influencing factors including secretor status and Lewis, the country of origin, and maternal physical status [35]. However, our study had several advantages. Firstly, focusing on a single population from China meant relatively smaller individual variations. Secondly, we allocated data available into five lactation stages and analyzed the dynamic changes throughout the lactation period. Lastly, the meta-analysis made it possible to offer the pooled individual concentrations at different lactation stages. 

The longitudinal changes in HMOs revealed in our study should be further validated. Inter-study variations are inherited weaknesses when combining data from different original research articles, as confirmed by our results where significant between-study heterogeneities were observed for most of the HMOs. Instruments, quantification methods, sample storage and preparation, and performance of experiments may all contribute to the variations. To further establish reference values of HMOs, standardized quantification methods should be developed. In addition, substantial publication biases were observed for some of the HMOs. Those HMOs are generally under-studied in Chinese population, which calls for future investigation. 

## 5. Conclusions

Our study represents the first systematic review that summarizes HMOs in Chinese population. We demonstrated that the levels of 3′-FL increase throughout lactation, whereas the levels of 6′-SL, LNT, and LNnT decrease with the progression of lactation. The longitudinal changes in HMOs during lactation suggest future directions investigating the functions and metabolism of HMOs, such as a direct comparison of 2′-FL and 3′-FL in their metabolism in infant bodies and formally validating the donation of sialic acid moieties by sialylated HMOs such as 6′-SL, 3′-SL, and DSLNT to the synaptogenesis process in brain development. 

## Figures and Tables

**Figure 1 nutrients-13-02912-f001:**
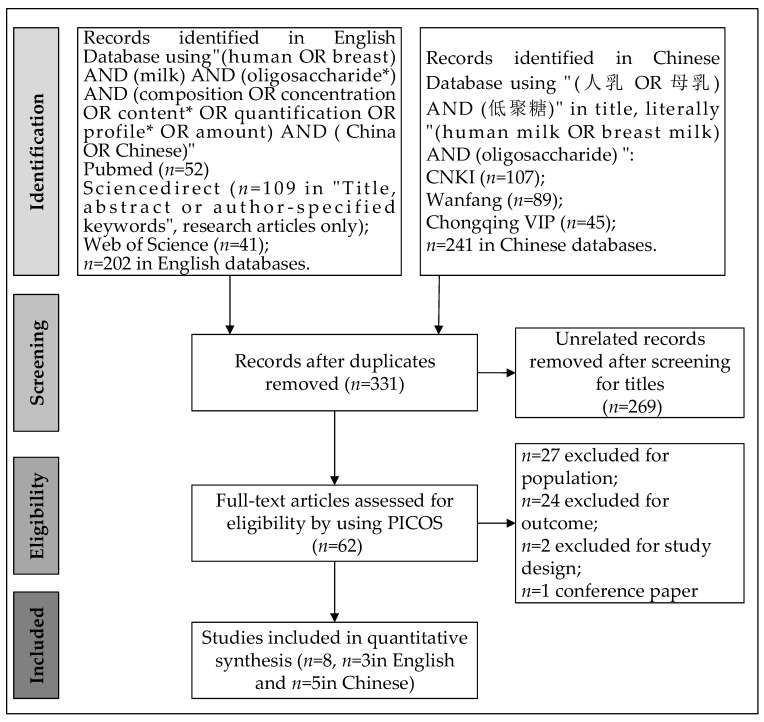
Flow chart of literature searching and screening.

**Figure 2 nutrients-13-02912-f002:**
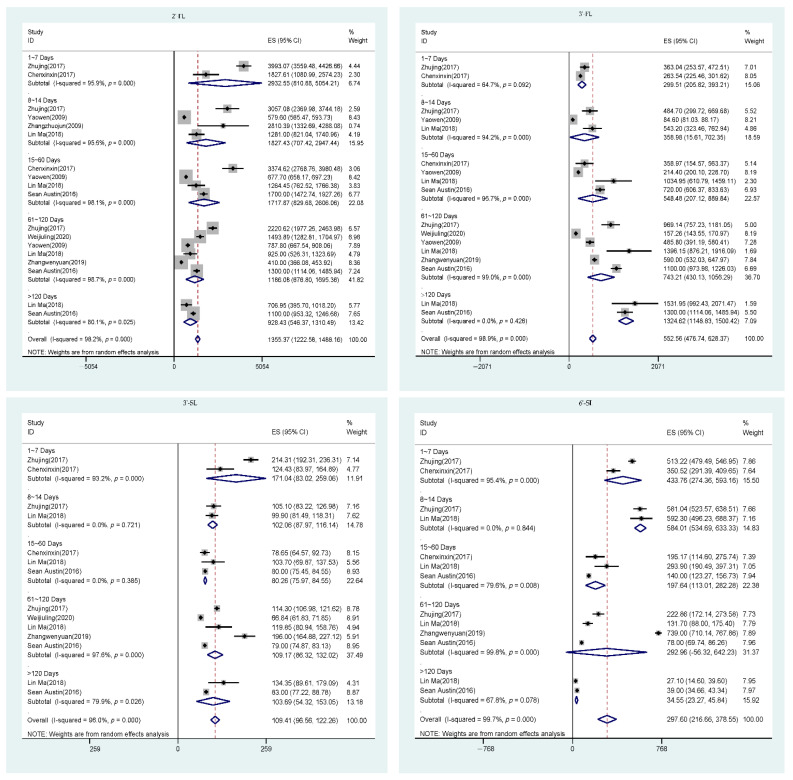
Forest plot of comparison of HMO concentrations.

**Figure 3 nutrients-13-02912-f003:**
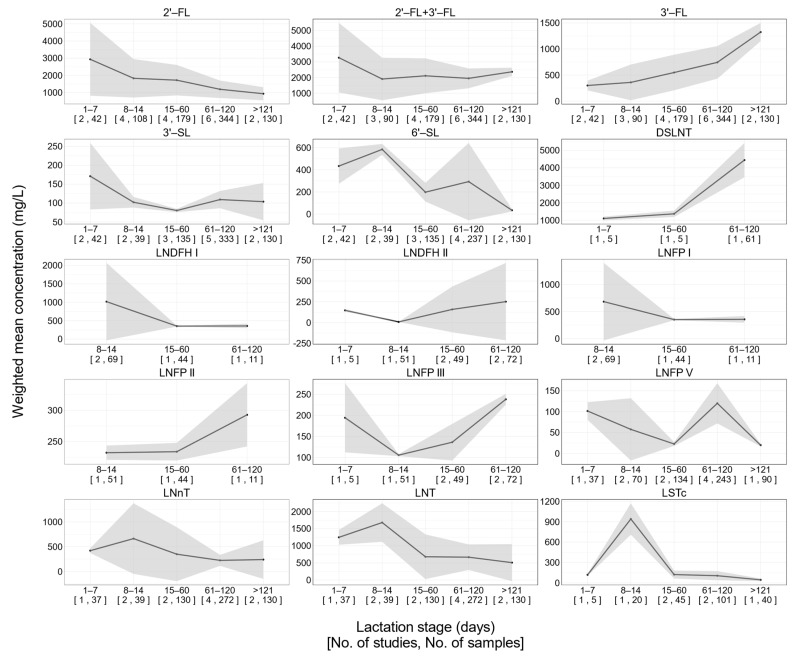
Longitudinal changes in individual HMOs.

**Figure 4 nutrients-13-02912-f004:**
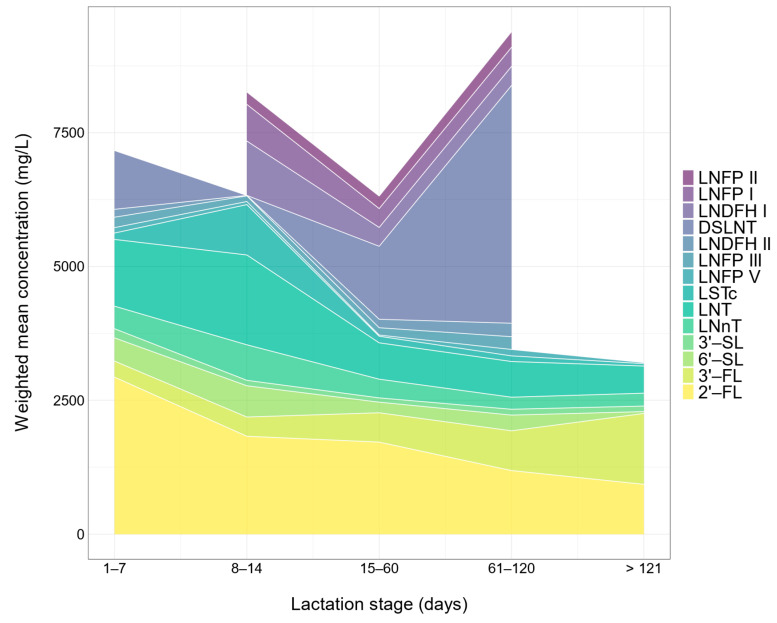
Longitudinal changes in HMO composition.

**Table 1 nutrients-13-02912-t001:** Inclusion and exclusion criteria for article selection.

Reference	Period of Investigation	Site	Collection Time (Days)	Number of Mothers	Milk Samples Collected	Term/Preterm	Secretor Status	HMOs Determination Method	Language	Measurement Unit
[4]	October 2011–February 2012	Beijing Guangzhou Suzhou	5–11 12–30 31–60 61–120 121–240	446	88 88 90 90 90	Full term	NR ^1^	UHPLC	English	mg/L
[14]	March–December 2018	Shanghai	8–14	30	27	Full term	FUT2(AA) FUT2(AT) FUT2(TT)	HPLC	Chinese	mg/L
[29]	NR	Guangzhou	14, 30, 60, 90, 120, 180, 240	NR	20	Full term	Non-secretor phenotype (37%)	HPLC–MS	English	mg/L
[30]	NR	Beijing	3 and 20	5	5 5	NR	NR	UPLC–QqQ-MS	Chinese	mg/L
[31]	NR	Nanjing Qiqihar	0–7; 8–15; 16–180	NR	102	NR	NR	UHPLC–FLD	Chinese	μg/g
[32]	Fedruary–May 2018	Beijing, Guangzhou, Nanjing, Zhengzhou, Mudanjiang, Chengdu	90 ± 15	96	10 6 20 20 20 20	Full term	NR	UHPLC–FLD	Chinese	μg/mL
[33]	March–July 2008	Shanghai	8–14 15–21 22–28	51	51 44 11	Full term	Le(a−, b+): 56.8% Le(a−, b−): 19.6%; Le(a+, b−): 23.6%	HPLC	Chinese	mg/L
[34]	NR	Beijing	31–180	61	61	Full term	NR	UPLC–tandem MS	English	mg/mL

^1^ NR, not reported.

**Table 2 nutrients-13-02912-t002:** Results of subgroup heterogeneity analysis.

	1–7 Days		8–14 Days		15–60 Days		61–120 Days		>121 Days		*Q*	*df*	*p*	*I* ^2^
	Mean	SE	Mean	SE	Mean	SE	Mean	SE	Mean	SE				
2′-FL	2932.5	1082.5	1827.4	571.4	1717.9	453.2	1186.1	259.8	928.4	194.9	7.0	4	0.137	42.7
3′-FL	299.5	47.8	359.0	175.2	548.5	174.2	743.2	159.7	1324.6	89.7	104.5	4	1.1 × 10^−21^	96.2
2′-FL + 3′-FL	3265.1	1131.7	1905.0	691.7	2111.0	567.6	1947.5	322.2	2365.4	135.7	2.6	4	0.629	0.0
3′-SL	171.0	44.9	102.1	7.2	80.3	2.2	109.2	11.7	103.7	25.2	18.1	4	0.001	78.0
6′-SL	433.8	81.3	584.0	25.2	197.6	43.2	293.0	178.2	34.6	5.8	484.9	4	1.2 ×10^−103^	99.2
DSLNT	1098.7	53.9	NA	NA	1363.8	92.7	4443.0	501.6	NA	NA	48.4	2	3.1 × 10^−11^	95.9
LNDFH I	NA	NA	1015.9	535.1	352.2	6.6	356.8	31.5	NA	NA	1.6	2	0.459	0.0
LNDFH II	147.3	8.1	8.9	0.3	158.6	140.0	251.8	236.6	NA	NA	290.7	3	1.0 × 10^−62^	99.0
LNFP I	NA	NA	684.4	365.3	352.2	6.6	356.8	31.5	NA	NA	0.8	2	0.655	0.0
LNFP II	NA	NA	232.4	5.8	234.0	7.2	292.7	25.7	NA	NA	5.3	2	0.072	62.0
LNFP III	194.3	41.9	105.6	1.3	136.2	22.2	238.1	6.6	NA	NA	393.2	3	6.7 × 10^−85^	99.2
LNFP V	101.6	10.7	57.7	38.0	22.8	2.4	120.0	24.5	20.0	1.6	73.5	4	4.1 × 10^−15^	94.6
LNnT	421.1	25.2	663.6	363.2	349.6	277.4	224.9	56.3	240.6	198.2	11.3	4	0.024	64.5
LNT	1247.6	110.4	1678.9	287.1	679.4	333.1	667.4	190.5	507.7	276.1	17.1	4	0.002	76.6
LSTc	118.9	7.8	940.9	118.1	122.5	29.9	105.7	34.4	44.8	9.9	87.3	4	4.9 × 10^−18^	95.4

Means are presented as weighted means calculated from included studies. *Q*: Cochran’s *Q* statistic which follows a chi-square distribution with df degrees of freedom; *df*: degrees of freedom; *I^2^*: Higgins and Thompson’s *I*^2^. NA: data not available.

## Data Availability

Not applicable.

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
