# Peer review of "Dynamic Changes in Human Milk Oligosaccharides in Chinese Population: A Systematic Review and Meta-Analysis"

_nutrients, 2021, doi:10.3390/nu13092912_

Round 1
Reviewer 1 Report
This manuscript by Zhou et al. describes a systematic review and meta-analysis of 8 studies of milk HMOs in Chinese populations. The rationale for this study is clearly articulated and the manuscript is generally well-written. There are a few areas that require clarification.
- A key concern is that measurements of several HMOs are only available for some time points from studies with very low number of participants. This is an issue given the evidence for relatively common genetic variants (such as at the Se gene) having a substantial impact on the levels of some HMOs, so it is unclear if these small samples (eg, n=5 for DSNLT at 1-7 and 15-60 days, see Figure 3) are comparable to the larger studies. This is problematic for both the discussion points on DSLNT levels increasing over time (since it is higher in a different, larger population, rather than the one used for the earlier time points) and on the findings of total HMO levels not decreasing over time (since the levels of some of the HMOs are based on an extremely small samples). It is important that the authors re-write the relevant parts of the discussion and conclusion to more accurately reflect the strength of evidence from this meta-analysis given this major limitation.
- One of the motivations for this study is the lack of Chinese populations included in a previous review. While it is stated in the discussion that the levels of HMOs cannot be directly compared between this study and the previous one, it would be of considerable interest to compare similarities or differences in relative HMO levels observed in the Chinese populations compared to the non-Chinese studies.
- For the discussion of DSLNT potentially being important for neurodevelopment due to its sialic acid moieties, would you be able to clarify what evidence there is for DSLNT or HMOs more broadly being taken up into the circulation? Earlier, the role of HMOs as probiotics is introduced, but this section on neurodevelopment would be strengthened by briefly summarising what existing evidence there is for the relationship between HMO levels maternal milk and offspring blood.
- In the discussion about the change of total concentration of HMOs over time, it is not clear how the method of calculating this differs from this manuscript ('combining the concentrations of all HMOs whose data were available') from the previous studies ('pooling the concentrations of individual HMOs'). Could this be rephrased/clarified?|
- While the manuscript is generally well-written, it would benefit from another proof-read to fix minor grammatical errors
Reviewer 2 Report
Overall, the manuscript is well organized and written. The review is informative and summarized the HMO characters in Chinese mothers.
If the authors could conclude some features or differences between Chinese and the population in the western world, the review will be more interesting.
